# Atmospheric isoprene measurements reveal larger-than-expected Southern Ocean emissions

Valerio Ferracci [1,7,13] ✉, James Weber [2,8,13] ✉, Conor G. Bolas[3,9], Andrew D. Robinson[3,10], Fiona Tummon[4], Pablo Rodríguez-Ros [5,11], Pau Cortés-Greus[5], Andrea Baccarini [6,12], Roderic L. Jones[3], Martí Galí [5], Rafel Simó [5], Julia Schmale [6] & Neil. R. P. Harris [1]

Isoprene is a key trace component of the atmosphere emitted by vegetation and other organisms. It is highly reactive and can impact atmospheric composition and climate by affecting the greenhouse gases ozone and methane and secondary organic aerosol formation. Marine fluxes are poorly constrained due to the paucity of long-term measurements; this in turn limits our understanding of isoprene cycling in the ocean. Here we present the analysis of isoprene concentrations in the atmosphere measured across the Southern Ocean over 4 months in the summertime. Some of the highest concentrations (>500 ppt) originated from the marginal ice zone in the Ross and Amundsen seas, indicating the marginal ice zone is a significant source of isoprene at high latitudes. Using the United Kingdom Earth System Model we show that current estimates of sea-to-air isoprene fluxes underestimate observed isoprene by a factor >20. A daytime source of isoprene is required to reconcile models with observations. The model presented here suggests such an increase in isoprene emissions would lead to >8% decrease in the hydroxyl radical in regions of the Southern Ocean, with implications for our understanding of atmospheric oxidation and composition in remote environments, often used as proxies for the pre-industrial atmosphere.

Emissions from the natural environment are a key component in the exchange of volatile organic compounds (VOCs) between the Earth's surface and the atmosphere. Isoprene (2-methyl-1,3-butadiene, $C_5H_8$) is estimated to account for half of all non-methane VOC emissions on the planet (~400–600 TgC year⁻¹)[1]. Whilst terrestrial sources account for the majority of the global isoprene budget, marine emissions have also been observed[2]. These are important in the relatively pristine marine air as they affect atmospheric composition by altering the oxidative capacity as well as climate via secondary organic aerosol (SOA) and cloud formation[3,4].

[1]Cranfield Environment Centre, Cranfield University, College Road, Cranfield, UK. [2]School of Biosciences, University of Sheffield, Sheffield, UK. [3]Department of Chemistry, University of Cambridge, Lensfield Road, Cambridge, UK. [4]Swiss Federal Office for Meteorology and Climatology MeteoSwiss, Payerne, Switzerland. [5]Institut de Ciències del Mar (ICM-CSIC), Barcelona, Catalonia, Spain. [6]Extreme Environments Research Laboratory, École Polytechnique Fédérale de Lausanne, Lausanne, Switzerland. [7]Present address: National Physical Laboratory, Hampton Road, Teddington, UK. [8]Present address: Dept of Meteorology, University of Reading, Reading, UK. [9]Present address: ITOPF, Old Broad Street, London, UK. [10]Present address: Schlumberger Cambridge Research, Madingley Road, Cambridge, UK. [11]Present address: Marilles Foundation, Bisbe Perelló, Palma, Mallorca, Spain. [12]Present address: Laboratory of atmospheric processes and their impact, École Polytechnique Fédérale de Lausanne, Lausanne, Switzerland. [13]These authors contributed equally: Valerio Ferracci, James Weber. ✉e-mail: v.ferracci@cranfield.ac.uk; j.m.weber@reading.ac.uk

Marine isoprene is thought to be primarily produced by phytoplankton in response to environmental stimuli (*e.g.*, temperature, solar radiation). Once dissolved within the surface layer of the ocean, it can undergo exchange with the atmosphere, be consumed by in-water processes or be transported deeper into the ocean[5]. In the atmosphere, the dominant fate of isoprene is reaction with the hydroxyl radical (OH) and, to a lesser extent, ozone ($O_3$) and the nitrate radical ($NO_3$), leading to less volatile oxidation products that can partition to the condensed phase or be further oxidised. The magnitude of global marine isoprene emissions is highly uncertain due to our incomplete understanding of its cycling within the ocean surface layers: for instance, recent studies suggest that chemical and biological isoprene consumption in the surface ocean may be a sink as important as exchange to the atmosphere[5]. While marine emissions have long been thought to arise primarily from biogenic sources, recent studies suggested that abiotic photochemical processes in the surface microlayer (SML) at the sea-air interface may also make a significant contribution to the overall marine isoprene budget[6]. However, many aspects of these processes are still unclear, and more studies are needed to establish the global distribution of this source and its magnitude relative to the known biogenic sources. Our limited understanding of the sources and sinks on marine isoprene results in discrepancies between bottom-up and top-down emission estimates spanning two orders of magnitude from ~300 GgC year$^{-1}$ [2,7–9] to ~11 TgC year$^{-1}$ [2,7].

Direct measurements of seawater-dissolved isoprene require in situ sampling, and are therefore limited to scientific expeditions. Estimates of sea-to-air isoprene fluxes from measured dissolved isoprene concentrations and wind speeds are even scarcer[10]. As an alternative, satellite retrievals of photosynthetically active pigments in ocean waters, such as chlorophyll-a, are often used as proxy for the presence of isoprene-emitting organisms and have allowed the development of emission schemes for dissolved isoprene concentrations and sea-to-air emissions[11,12].

The Southern Ocean (SO) is a known source of climate-relevant VOCs (*e.g.*, dimethyl sulphide) as a result of the richness of nutrients in its waters, which sustain varied populations of emitting organisms. The SO is uniquely important to understanding natural chemistry-climate processes in present-day pristine environments and it is often used as a proxy for pre-industrial environmental conditions[13,14]. An improved understanding of SO VOC emissions, and how they impact atmospheric composition (*e.g.*, via oxidation reactions) and climate (*e.g.*, via formation of SOA and cloud condensation nuclei) is therefore key to elucidating atmospheric composition and processes in the pre-industrial period[15]. Observations of ambient isoprene above the SO are few and far-between[10,16–20]. Generally, low atmospheric mixing ratios (<50 pmol mol$^{-1}$, or ppt) are found in regions of the SO away from landmasses and phytoplankton blooms, whilst higher values up to a few hundred ppt can be found near coastal areas and regions of high biological activity. Estimates from observations suggest low isoprene fluxes, in the order of tens of nmol m$^{-2}$ d$^{-1}$, equivalent to ~0.8-10 × 10$^{-14}$ kg m$^{-2}$ s$^{-1}$ [12].

We measured atmospheric isoprene across the SO during the Antarctic Circumnavigation Expedition (ACE) in the austral summer of 2016–17 (Fig. 1a). We reported unusually high concentrations in remote regions of the SO and investigate their sources. We then evaluated current emission parameterisations by comparing airborne concentrations simulated by the United Kingdom Earth System Model (UKESM1) using a range of different marine isoprene emission inputs, to observed isoprene concentrations. Finally, we quantified the impact of these emissions on OH, the key tropospheric oxidant.

## Results
### Isoprene observations and source regions

Ambient isoprene mixing ratios from ACE are shown in Fig. 1b and S1. Airborne isoprene was available for the first third of Leg 1 (from Cape Town to the Crozet Islands, 35–47°S), for most of Leg 2 (from Hobart to Punta Arenas, 43–72°S), and only for the final few days of Leg 3 (approaching Cape Town, 34–46°S). The mean isoprene mixing ratios were 42, 33, 47, 38 ppt for the whole campaign, Leg 1, Leg 2, and Leg 3, respectively. These values are largely in agreement with previous measurements in the Atlantic sector of the SO (53 ± 34 ppt)[10]. Particularly high mixing ratios (up to 1200 ppt) were observed at high

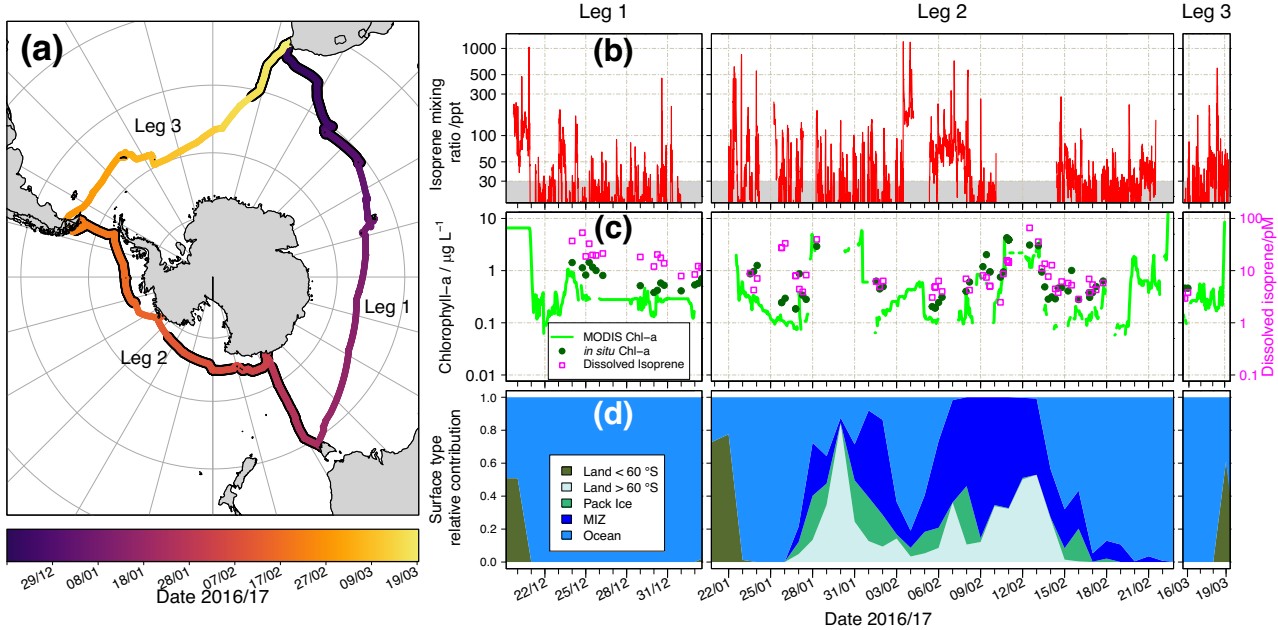

**Fig. 1 | Summary of isoprene measurements and related variables during the Antarctic Circumnavigation Expedition (ACE). a** Ship track for the ACE campaign. Sections of the track with a black border indicate where measurements of atmospheric isoprene are available; (**b–d**) Time series for: observed atmospheric isoprene mixing ratios (**b**, note the log scale), with the grey area indicating data below instrument LOD (30 ppt); in situ chlorophyll-a and dissolved isoprene along with chlorophyll-a from MODIS-Aqua (**c**); daily mean relative contribution of surface types along the air mass back trajectories (**d**).

latitudes during Leg 2 as the ACE vessel traversed the Ross Sea (3–9 Feb 2017), and were accompanied by a marked shift in baseline from below the instrument limit of detection (LOD, 30 ppt) to ~70 ppt. We discounted interferences from the exhaust plume of the ship (Fig. S2 and Supplementary Information) and found that contributions from land sources were limited to the vicinity of mid-latitude landmasses (Fig. 1d). These elevated abundances at high latitudes did not correspond to a particularly shallow boundary layer (Fig. S1c), but they did at times correspond to high wind speeds, potentially indicating stronger sea-to-air exchange if dissolved isoprene was available along the air mass back-trajectory.

At mid-latitudes (30–50°S), particularly during Leg 1, atmospheric isoprene and chlorophyll-a (Fig. 1c) followed similar decreasing trends as the ship travelled away from the coast (see Supplementary Information and Table S1). However, the two decoupled in Leg 2, especially at high latitudes (>70°S), where the elevated isoprene mixing ratios and baseline enhancement in Leg 2 did not correspond to high dissolved isoprene or chlorophyll-a concentrations (see Table S1). When we consider the variation in isoprene atmospheric lifetime, $\tau_{isop}$, with latitude (Fig S3) we find that $\tau_{isop}$ was relatively short at mid-latitudes (1–5 hours), indicating that the measurement footprint was influenced by local sources only. At higher Southern latitudes such as those in Leg 2, $\tau_{isop}$ was much longer (>10 hours), indicating that emissions further afield influenced the observed concentrations. An analysis of the air mass history was therefore necessary to understand the sources in this region. Figure 1d illustrates the relative contribution of different surface types to each air mass history along the ACE track. The high isoprene abundances observed half-way through Leg 2 (03-09 Feb 2017) exhibited a significant contribution from the marginal ice zone (MIZ).

Upon closer inspection, these high concentrations were associated with air masses travelling over areas of high biological activity in the Ross Sea, as indicated by high values of chlorophyll-a (Fig. 2a). These regions also overlap with areas of marginal ice (Fig. S4), indicating that the enhanced chlorophyll-a was the result of a phytoplankton bloom following sea ice melt. The concentrations of isoprene in air parcels originating predominantly from the marginal ice zone (MIZ) were typically higher than those from open waters, and second in magnitude only to those originating from temperate land (Fig. 2b). This is consistent with previous observations in the Canadian Arctic[21,22] and the Antarctic peninsula[23], where unexpectedly high

concentrations of biogenic VOCs in water were measured at the MIZ and linked to phytoplankton blooms driven by the penetration of sunlight into the water as the ice retreated. Our observations show that these blooms are also a significant source of isoprene in Antarctic waters.

## Model - observation comparisons

To assess the ability of models to capture the observed airborne isoprene concentrations and investigate their wider atmospheric implications, simulations were performed using the state-of-the-art UKESM1 with varying prescribed emissions of terrestrial and marine isoprene (Table 1; Fig. S5). Specifically, simulations were performed to (i) identify the extent to which marine observations could be explained by the transport of isoprene from terrestrial sources; (ii) explore model performance when using marine isoprene emissions derived from the satellite-derived parameterisation ISOREMS, which was developed from ACE measurements and other observations in the SO[12]; and (iii) investigate the impact that applying diurnal emission scaling and adding co-located emission of other VOCs (which would suppress the main sink of isoprene, OH) would have on simulated isoprene concentrations.

The use of atmospheric nudging (Methods) ensured the simulated meteorology was as close to that experienced during the ACE cruise as possible. Given the importance of wind speed and direction for the transport of isoprene, we verified that the horizontal wind fields in the model were similar to those in ACE (Figs. S6–9).

The absence of simulated isoprene away from land in TI_base (Fig. 3a) demonstrates that terrestrial emissions alone are unable to account for the isoprene concentrations measured at sea. This is further reinforced when a doubling of terrestrial emissions below 30°S (TI_2x_30S), which represents an upper bound uncertainty in isoprene emissions[1,4], yielded the same results. The influence of terrestrial emissions is limited to the vicinity of temperate coastal areas, as also indicated by the air mass trajectory analysis (Fig. 1d).

When marine isoprene emissions based on the ISOREMS parameterisation are included in model simulations (TI_MI_MEAN), isoprene concentrations along the ship track increase. Direct comparison to TI_base and the use of separate terrestrial and marine isoprene model tracers (Methods) reveals this increase to be driven solely by marine isoprene. However, even with the addition of these marine emissions, modelled isoprene is still strongly low biased (model/

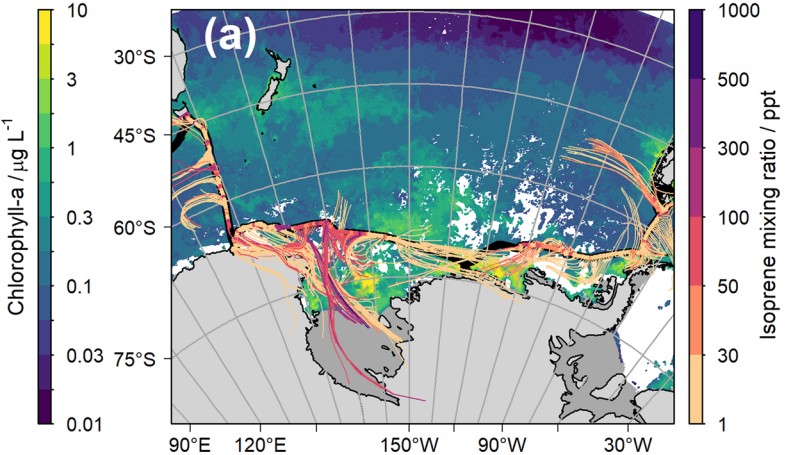

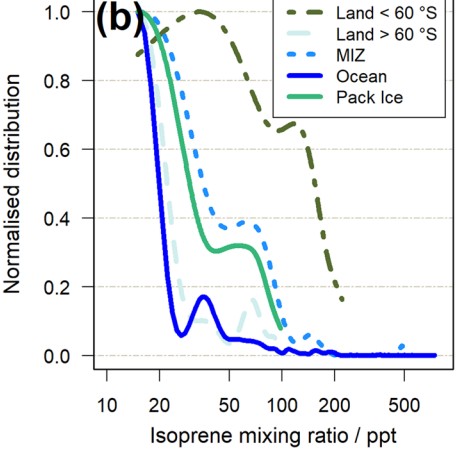

**Fig. 2 | Back-trajectory analysis and contribution from different surface types to the observed isoprene. a** Map of composite chlorophyll-a concentration (strip legend on the left) from MODIS-Aqua from 25 Jan to 17 Feb 2017, corresponding to the second leg of the Antarctic Circumnavigation Expedition (ACE). The ACE track is shown as a black solid line, with overlaid atmospheric isoprene mixing ratios (strip legend on the right). The back-trajectories, shown as thin coloured lines, are

adjusted for the isoprene lifetime at each point along the cruise track, with a maximum allowed lifetime of 48 h. The colour of each back-trajectory is the same as the isoprene mixing ratio at the point along the cruise track from which it originates. Ice shelves are shown as dark grey areas. See also Fig. S4 for a version of this map showing Sea Ice Fraction. **b** Normalised distributions of isoprene mixing ratios influenced by different surface types.

**Table 1 | Model simulations and corresponding prescribed isoprene emissions**

| Simulation | Isoprene emissions | Ratio of simulated / observed Isop concentration (median$_{25th\ percentile}^{75th\ percentile}$) |
|---|---|---|
| TI_base | Terrestrial only | $<10^{-9}$ |
| TI_2x_30S | Terrestrial only with emissions below 30°S doubled. | $<10^{-9}$ |
| TI_MI_MEAN | Terrestrial emissions and marine emissions calculated following ISOREMS parameterisation[12] | $0.025_{0.008}^{0.062}$ |
| TI_MI_20x | As TI_MI_MEAN but marine emissions scaled by 20. | $0.490_{0.166}^{1.153}$ |
| TI_MI_MEAN_D | As TI_MI_MEAN but with a diurnal scaling applied to marine isoprene emissions. | $0.019_{0.006}^{0.046}$ |
| TI_MI_20x_D | As TI_MI_20x but with a diurnal scaling applied to marine isoprene emissions. | $0.368_{0.146}^{0.838}$ |
| TI_MI_MEAN_D_sink | As TI_MI_MEAN_D with an extra species, "OH-sink", which has (non-diurnally varying) emissions equal to those of marine isoprene. OH-sink has a rate constant half that of isoprene + OH and does not regenerate any OH. | $0.018_{0.006}^{0.045}$ |
| TI_MI_MEAN_D_sink_100x | As TI_MI_MEAN_D_sink but OH-sink emissions are 100x those of marine isoprene. | $0.021_{0.008}^{0.053}$ |

All simulations were performed for the period Dec 2016 - Mar 2017 (inclusive). Terrestrial emissions were taken from the 2001–2010 MEGAN-MACC climatology (Methods) and featured a diurnal scaling. Marine isoprene emissions did not have a diurnal scaling unless otherwise stated.

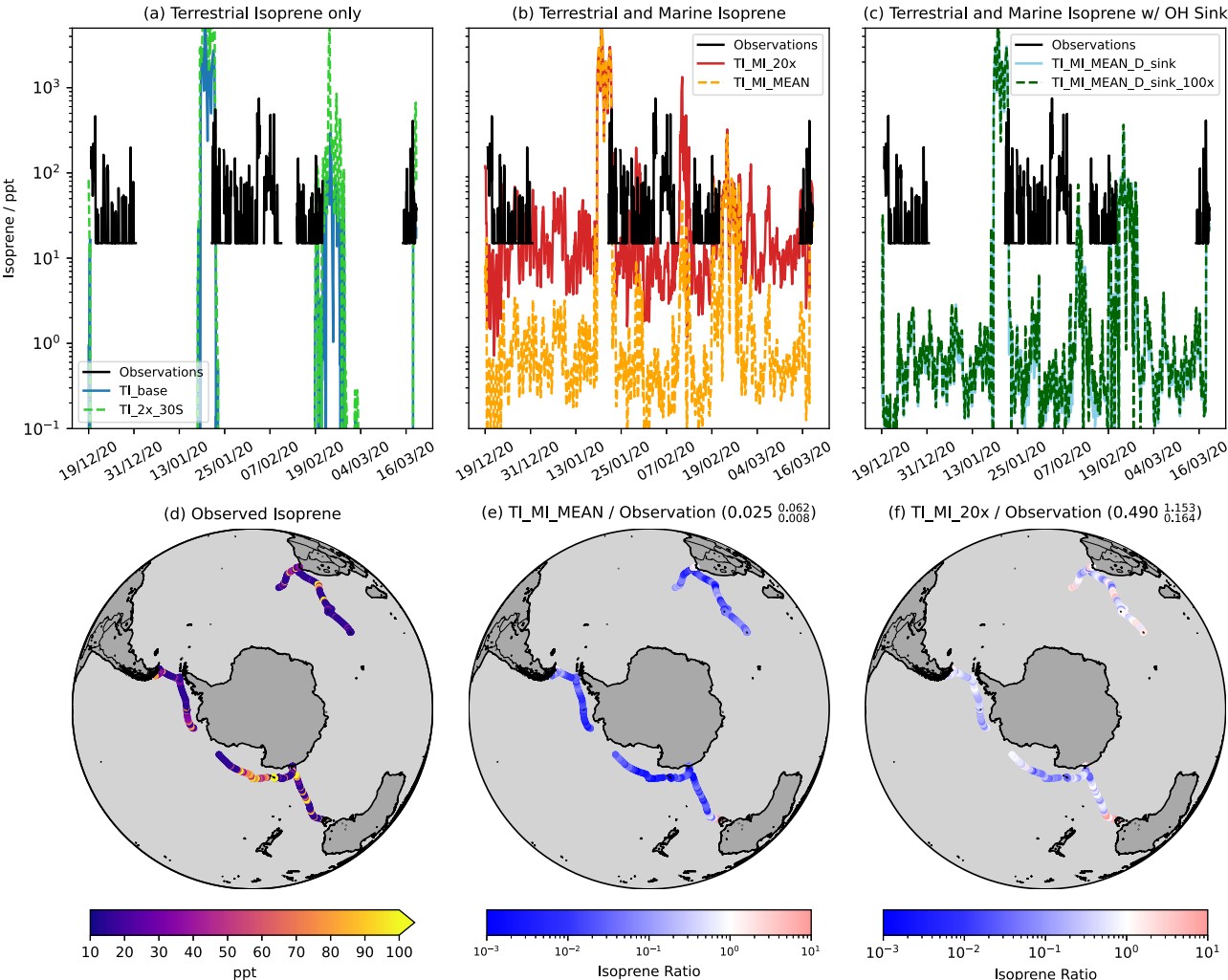

**Fig. 3 | Comparison of observed isoprene with modelled concentrations.** Observed isoprene along ship track (**a–d**) and modelled isoprene from simulations with (**a**) terrestrial isoprene emissions only (TI_base) and terrestrial isoprene emissions doubled below 30°S (TI_2x_30S), (**b**) terrestrial emissions with marine emissions calculated following the mean parameterisation from Rodríguez-Ros[12] (TI_MI_MEAN) and 20x these marine isoprene emissions (TI_MI_20x) and (**c**) terrestrial and marine emissions as TI_MI_MEAN along with an additional sink of the hydroxyl radical, OH (details in Table 1), the emissions of which are equal to (TI_MI_MEAN_D_sink) and 100 times (TI_MI_MEAN_D_sink_100x) those of marine isoprene. Ratio of modelled isoprene emissions from (**e**) TI_MI_MEAN and (**f**) TI_MI_20x to observed isoprene. Values in (**e**) and (**f**) show median$_{25th\ percentile}^{75th\ percentile}$ of ratio across the whole campaign.

observation ratio median$_{25th\ percentile}^{75th\ percentile}$ = 0.025$_{0.008}^{0.062}$) (Fig. 3b, e). In an effort to address this low bias, we scaled these marine isoprene emissions by a factor of 20 (TI_MI_20x). This improved the model performance but it remained low-biased by a factor of ~2 (model/observation ratio median$_{25th\ percentile}^{75th\ percentile}$ = 0.490$_{0.164}^{1.153}$) (Fig. 3b, f). Applying the same diurnal cycle to marine isoprene emissions as that applied to terrestrial isoprene emissions in UKESM1 also has little effect on the model performance (Fig. S10).

A potential driver of this low bias is the suppression of OH by other VOCs present in the atmosphere but not included in the simulations[24,25]. However, we ruled this out with simulations where a dummy species, which reacts with OH at half the rate of OH+isoprene and does not regenerate OH, is included with the same (TI_MI_MEAN_D_sink) and 100 times (TI_MI_MEAN_D_sink_100x) the emissions of marine isoprene. The former scenario has a negligible impact on OH, while the latter yields OH reductions of 9–15% in much of the SO, exceeding 15% off the coast of South America, relative to TI_MI_MEAN_D (Fig. S11).

This suppression of modelled OH does affect isoprene: along the ship track OH is on average 16.4% lower in TI_MI_MEAN_D_sink_100x than in TI_MI_MEAN_D, with isoprene concentrations 16.7% higher, but this difference (mean 0.9 ppt) is still 2 orders of magnitude lower than the model-observations bias (Fig. 3c). Therefore, a much larger model OH bias would be needed to reconcile the model with observations. However, the simulated mean OH concentrations for the December–March period ($\geq 4 \times 10^5\ cm^{-3}$ for Legs 1 and 3 and $2$–$3 \times 10^5\ cm^{-3}$ for most of Leg 2 (Fig. S1)) compare reasonably well to the coastal Antarctic OH measurements of $1.1 \times 10^5\ cm^{-3}$ (Feb 1994: Palmer Station) and $3.9 \times 10^5\ cm^{-3}$ (Jan–Feb 2005; Halley Station)[26]. This suggests that variation in OH is not the main driver of the model low bias, which is likely caused by underestimation of emissions and/or biases in the simulation of vertical transport.

A lack of observational data on updraught velocity from the ACE ship prevents evaluation of the model's vertical transport. To consider one moderate and one extreme case of model overestimation of vertical transport, we calculate the simulated isoprene concentration that would occur if all the isoprene in the lowest 5 and 11 model levels (extending to 280 m and ~1000 m above the surface, respectively) were compressed into the lowest model level (~35 m in height) in the case of TI_MI_MEAN along the ship track. The resulting simulated isoprene is still low biased in both cases considered (model/obs ratio median$_{25th\ percentile}^{75th\ percentile}$ = 0.151$_{0.048}^{0.364}$ and 0.313$_{0.122}^{0.755}$ for the 5 and 11 level compression respectively; Fig. S13), suggesting that the major driver of the model-observation discrepancy is differences in emissions.

## Discussion

The ACE measurements of atmospheric isoprene are consistent with the, albeit sparse, previous observations in the SO, with mean values in the order of tens of ppt and maxima in the order of hundreds of ppt[10,17–20]. Therefore, and given the limited data available, our measurements are representative of the SO and not unusually high for this region and time of year.

As discussed in the previous section, we systematically analyzed transport from land, vertical mixing and sinks (OH and presence of other VOCs), and ruled them out as the origin of the discrepancy between modelled and observed concentrations. Rather, the discrepancy must arise from a missing isoprene source. Analysis of the observed isoprene diel pattern provides clues on its potential origin.

Whilst isoprene biological production in water is linked to photosynthetic activity, previous studies in the open ocean observed either no significant diurnal pattern in the concentrations of seawater isoprene[11], or marginally (<10%) higher daytime concentrations[10]. This is potentially due to a large share of the production occurring at depths greater than 5 m, with the subsequent vertical mixing effectively smoothing out any diel signature[11]. Along with the lack of a

marked diurnal pattern in wind speeds (Fig. S14), this would lead to an isoprene flux to the atmosphere with no distinct diel variation (equivalent to TI_MI_20x), which would in turn yield a diurnal minimum in atmospheric isoprene as a result of daytime removal by OH (Fig. 4a). However, observations of marine-only air masses (Methods) show higher mixing ratios during daytime and lower at night (Fig. 4a, b and S15), even when latitudinal variations in hours of sunlight are considered. This is also the case when: (i) values below the instrument LOD are set to zero (Fig. S16); (ii) values are normalised to the daily mean isoprene (Fig. S17)[10]; (iii) the median is used instead of the mean (Fig. S18). A similar diel pattern was also previously observed in the SO[10,17]. It follows that the flux of isoprene to the atmosphere must be higher during the day than at night if a daytime maximum is to be observed. This is simulated in the TI_MI_20X_D model run, in which a diel cycle with a daytime maximum is imposed on the emissions, resulting in higher daytime modelled concentrations (Fig. 4b).

Production at the surface microlayer (SML) from photochemistry of surfactants at the air-water interface has been proposed as an additional marine source of isoprene[27,28]. Not only would this source exhibit the temporal pattern needed to account for the observed diel variation in isoprene concentrations, but it would also provide emissions of the magnitude needed to better reconcile models with observations, as fluxes up to $0.5$-$20 \times 10^{-12}\ kg\ m^{-2}\ s^{-1}$ were reported in laboratory studies of photosensitised SML[27–29]. Brüggeman et al.[6] showed that the SML would not be stable in the SO between 40 and 60°S because of strong winds, while it would be near coastal Antarctica. This is consistent with our observations (Fig. S15): the diel cycle was more pronounced during Leg 2 (wind speed <10 m s$^{-1}$, Fig. S14), closest to coastal Antarctica, than in Leg 1 (wind speed ≥10 m s$^{-1}$, Fig. S14). This mechanism would be reinforced by ice melt, which would stabilise the ocean surface layer by the addition of buoyant freshwater. Ice melt might also supply precursors for photochemical isoprene production from sea ice microbial biomass. Together, these processes would make the surface layer at the MIZ a strong source of isoprene. We conclude that the observed atmospheric isoprene patterns result from a combination of biotic emissions from bulk seawater (equivalent to TI_MI_20x in Fig. 4a),and light-dependent SML emissions (equivalent to TI_MI_20x_D in Fig. 4b).

We show that present-day isoprene emissions in the SO need to be scaled up by at least a factor of 20 to reconcile our model with observations. While only 0.44% of global isoprene sources (Fig. 4c), the increased emissions across the SO impact the composition of the regional atmosphere. For the DJFM period considered, OH is reduced on average by 2.1% in the SO with regions off the eastern coast of South America and areas of phytoplankton blooms exhibiting decreases >8% (Fig. 4d). The pristine nature of the SO in the present day and, to an even greater extent, in the pre-industrial era will amplify the impact of such composition changes[30], with important effects on climatically-relevant processes including $SO_2$ oxidation, organic and sulphate aerosol formation, and cloud droplet activation.

We note that the DJFM period considered here, chosen to coincide with the ACE campaign and thus allow us to constrain marine isoprene emissions to observations, is the period in the year with the highest SO marine isoprene emissions[12]. Therefore, including marine isoprene emissions in this period will have the largest absolute impact on the hydroxyl radical and wider atmospheric composition. However, marine isoprene emissions are far from negligible in other parts of the year, even in the austral winter (2–3 GgC/month in June and July compared to 8 GgC/month in January[12]). The austral winter's much lower OH concentration, due to the lower primary production from ozone photolysis given the reduced intensity and duration of sunlight, means the relative change in OH with the inclusion of marine isoprene could be comparable to that in austral summer. To probe the impact of a 20-fold scaling for the annual mean change in surface OH, we ran a

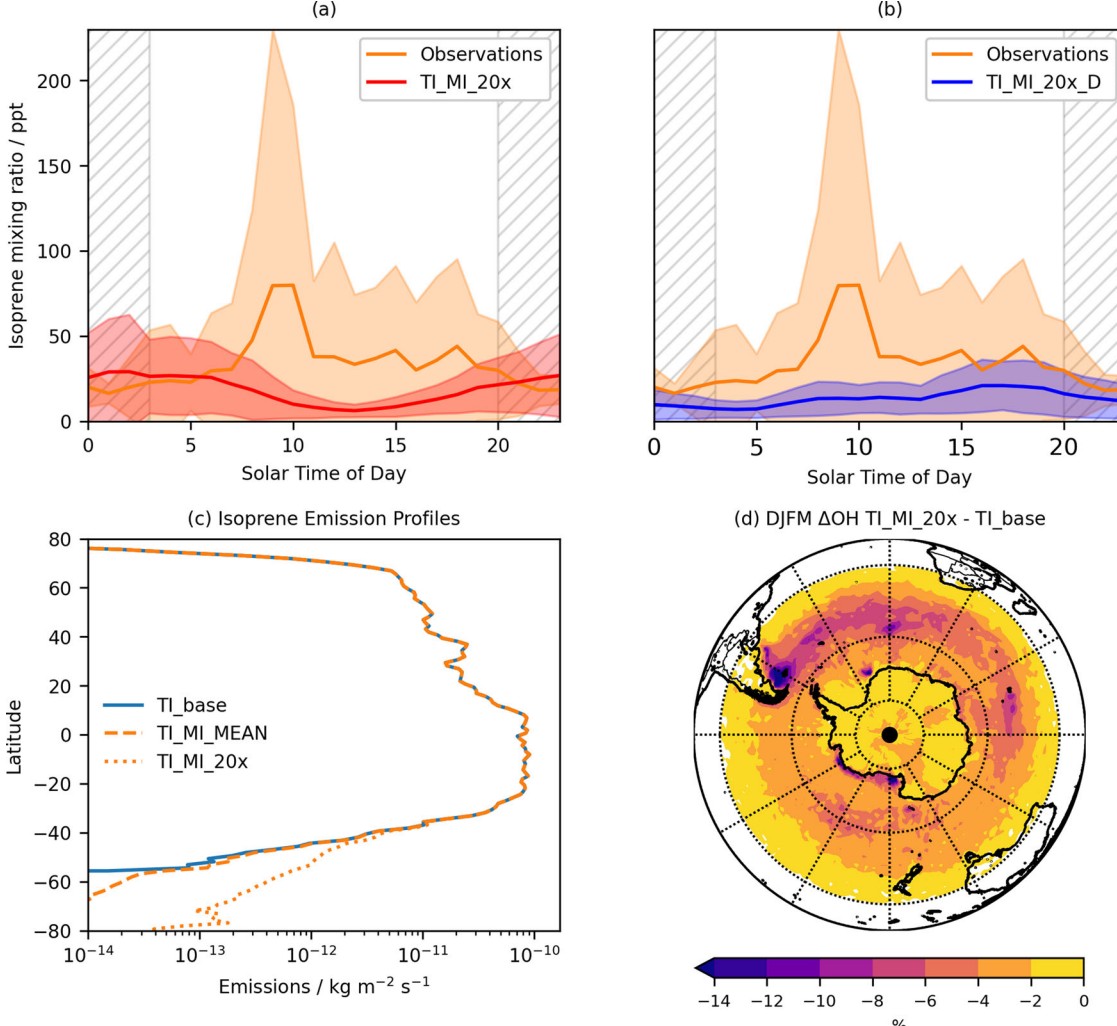

**Fig. 4 | Isoprene diel cycle analysis and atmospheric impacts of enhanced isoprene emissions.** Mean diel cycle of marine-only isoprene (orange line) along with the modelled diel cycles for (**a**) TI_MI_20x and (**b**) TI_MI_20x_D for the entire Antarctic Circumnavigation Expedition (ACE). The gray-hatched areas indicate nighttime (the length of which varies by ±1 h throughout the cruise); shaded areas represent 1 standard deviation above and below each 1-h mean. See Fig. S15 for the diel cycles of the different legs of ACE. The 0 solar time of day corresponds to 00:00 local time and 23 corresponds to 23:00 local time. **c** Latitudinal profile of annual mean isoprene emissions considered in this study, including terrestrial only emissions (TI_base, defined in Table 1), terrestrial and marine isoprene emissions with marine emissions taken directly from ISOREMS[12] (TI_MI_MEAN, also defined in Table 1), and terrestrial and marine isoprene emissions with marine emissions taken from our own calculations and scaled by a factor of 20 (TI_MI_20x, also defined in Table 1). **d** Percentage change in OH in lowest ~150 m between TI_MI_20x and TI_base for the months of December, January, February and March (DJFM).

further simulation with 20x marine isoprene emissions for a whole year and compared it to the annual mean of TI_base. The change in annual mean OH was 1.7% (compared to 2% for DJFM), with a similar spatial distribution to DJFM (Fig. S19), but we note the lack of observational constraints means the validity of a 20-fold scaling for the whole year is more questionable. Overall this highlights the need for further research into the atmospheric marine isoprene concentrations throughout the year in the SO so that a full understanding of their role in atmospheric composition, aerosol and climate change can be established.

Future changes to marine isoprene emissions will depend on a complex mix of factors including chlorophyll-a concentrations (predicted to increase under high (SSP3-7.0; +5.8%) and low (SSP1-2.6; +2.9%) warming scenarios in the SO, Fig. S20), sea-ice dynamics, SSTs and surface winds. Significant fluctuations in sea ice extent with a general decreasing trend (e.g., record low in 2023) are likely to have important implications for the MIZ and related isoprene emissions, and this work provides a baseline for this rapidly changing

environment against which future observations and models can be compared. These rapid changes indicate that any further measurements of baseline conditions have to be made urgently.

Our work shows how greater isoprene emissions in the SO have significant implications for our understanding of the atmospheric composition of both present-day remote environments and of the predicted pre-industrial baseline, with consequences for modelled pre-industrial-to-present-day changes in atmospheric composition and attendant radiative forcing.

In recent years, multiple studies have reported that isoprene and other VOCs are emitted in marine environments at higher rates than previously thought[24,25,31]. This could help explain the observed concentrations of VOC oxidation products and aerosol precursors, such as glyoxal and methylglyoxal, from marine sources[32,33], and help close the OH budget over marine regions[34]. Moreover, the hygroscopicity of cloud condensation nuclei in the SO has been found to be much lower than expected, pointing towards a significant role of organic aerosol species[13,35]. The higher-than-expected isoprene emissions in the region

are likely one explanation. Model simulations where isoprene is coupled explicitly to the organic aerosol scheme could be used to probe this in future work, with detailed comparison to observations.

Our work addresses the discrepancy between bottom-up and top-down estimates of marine isoprene emissions[8,9], indicating that seawater-derived (bottom-up) fluxes are insufficient to explain the concentrations of atmospheric isoprene in the SO during the austral summer, and that fluxes at least 20 times larger are needed. The diel cycles observed in this work strongly indicate that photochemical production of isoprene at the sea-air interface may be a significant contributor to the total isoprene budget. We recommend that future studies focus on direct flux measurements of isoprene, supported by measurements at the SML and in bulk water, to elucidate the sources to the atmosphere and fill the gaps in our current understanding of isoprene cycling in remote marine environments. This will in turn allow a more accurate representation of pre-industrial-to-present-day changes in atmospheric composition and oxidising capacity in Earth System models.

## Methods
### The ACE campaign
The Antarctic Circumnavigation Expedition (ACE) was a multidisciplinary campaign aboard the R/V Akademik Tryoshnikov, aiming to investigate the biogeochemistry of the Southern Ocean in unprecedented detail, with a wide suite of measurements spanning marine biology, atmospheric chemistry and meteorology, amongst other research areas.[13,36,37]

ACE was divided into three legs: Leg 1, from Cape Town (20 December 2016) to Hobart (18 January 2017); Leg 2, from Hobart (22 January 2017) to Punta Arenas (22 February 2017); Leg 3, from Punta Arenas (26 February 2017) to Cape Town (19 March 2017).

Ambient air was sampled from an inlet mounted on a container located on the second deck of the vessel at a height of approximately 15 m above the ocean surface[36]. The main inlet consisted of a 2 m, 1" OD stainless steel tube heated to 20 °C. Ambient air was sampled at 13 L min⁻¹, resulting in a residence time in the main inlet of <3 s. Mixing ratios of ambient isoprene were monitored using the iDirac, an autonomous portable gas chromatograph coupled with photoionisation detection (GC-PID)[38]. The iDirac sampled from the main air inlet via an ~ 50 cm, 1/16" OD, 0.04" ID SilcoNert2000-treated stainless steel tube, at a rate of 20 mL min⁻¹, resulting in a residence time of 1–2 s. The use of passivated surfaces minimised surface losses. The instrument was calibrated frequently (every 3–5 hours) using a gas mixture of isoprene in nitrogen prepared in-house and traceable to the National Physical Laboratory (NPL) primary standards[38]. Across the ACE campaign, the limit of detection for isoprene was 30 ppt (or pmol mol⁻¹) with 10% precision. Overall, just under 36% of the isoprene data recorded was above the LOD of the iDirac. In accordance with Hackenberg et al.[11], values below the LOD were set to half the LOD (15 ppt). All analyses of the isoprene data in this work incorporate this adjustment, unless stated otherwise.

A wealth of data was collected during the ACE campaign. The majority of these datasets are available in the ACE online repository[39]. The following were used for the analysis of ambient isoprene. Distance to land was calculated using the coordinates of the ship track at a 5 min resolution[40]. Ocean water samples were collected throughout the cruise and later analysed for a number of dissolved species, including trace gases (e.g., isoprene) and fluorometric chlorophyll-a[41] as described by Rodríguez-Ros et al.[42] Fluorometric chlorophyll-a was corrected according to Galí et al.[43] to allow comparison with satellite retrievals.

Wind speed data underwent a thorough correction to account for air flow distortion due to the ship's structure[44]. Further details on the correction can be found in Landwehr et al.[45] Trace gas concentrations, including ozone and carbon monoxide (CO), were taken from Schmale et al.[46] A number of meteorological parameters such air and sea surface temperature, and boundary layer height were obtained by interpolating the output from the ERA-5 reanalysis[47], available at 1 h temporal and 0.25° × 0.25° spatial resolution, onto the ACE cruise track[48]. Five-day air mass back trajectories were calculated using the Lagrangian analysis tool LAGRANTO[49,50].

### Satellite retrievals
Chlorophyll-a concentrations in water at a resolution of 0.0416° × 0.0416° were obtained from the Level-3 product of the Moderate Resolution Imaging Spectroradiometer (MODIS-Aqua[51], last accessed 23/03/2022). These were used to determine the concentration of chlorophyll-a along the ACE track (Figs. 1c, S1b). The presence of persistent cloud cover and sea ice gives rise to many gaps in the retrieved ocean chlorophyll-a over the region of interest, especially at high Southern latitudes. This was alleviated by using 8-day composite retrievals as well as by averaging all pixels in a 9 × 9 array centred on each position along the ACE track. However, if chlorophyll-a concentrations were not available for ≥ 50% of the pixels in each array, those points were excluded from any subsequent analysis.

Sea ice concentration (SIC) fraction from NOAA/NSIDC[52] was used for the analysis of surface type contribution and to differentiate between the marginal ice zone (MIZ) and the more consolidated pack ice.

### Isoprene lifetime calculations
The chemical fate of ambient isoprene is primarily to undergo reaction with the hydroxyl radical (OH) and, to a lesser extent, with ozone and the nitrate radical (NO₃)[53]. Abundances of all three oxidants are needed to establish the atmospheric lifetime of isoprene at a given location. Ozone concentrations were monitored throughout ACE, while those of OH and NO₃ were not measured. For the analysis presented here, modelled OH and NO₃ abundances were taken from two global models: the UM-UKCA model[54] and the CAMS reanalysis of atmospheric composition[55], which incorporates meteorological variables from the ERA-5 reanalysis. Isoprene lifetime with respect to atmospheric oxidation, $\tau_{isop}$, was calculated as:

$$\tau_{isop} = 1/\{k(OH + isop)[OH] + k(O_3 + isop)[O_3] + k(NO_3 + isop)[NO_3]\}$$

(1)

where $k$ indicates the rate coefficient of the reactions of isoprene with OH, ozone and NO₃ (in units of cm³ molecule⁻¹ s⁻¹) and [OH], [O₃] and [NO₃] are the number densities of the hydroxyl radical, ozone and the nitrate radical, respectively (in units of molecules cm⁻³). Rate coefficients were taken from the IUPAC kinetic database[56]. Air temperature from on-board measurements was used to calculate the temperature dependence of the rate coefficients for each reaction. $\tau_{isop}$ was used to adjust the length of each back-trajectory, so that effectively the back-trajectory for an air parcel at night-time (long $\tau_{isop}$) will stretch further than one at day-time (short $\tau_{isop}$). It is worth noting that using a fixed $\tau_{isop}$ of 2–3 h (typical at mid-latitudes) would limit all trajectories to the immediate vicinity of the ACE track, missing on the influences of different surface types (e.g., marginal ice), as illustrated in Fig. 2 and S4.

### Surface-type contribution
The points in the back trajectory for each isoprene measurement were assigned to a particular surface type depending on whether the air mass travelled above land, ocean or sea ice. A land mask from the ECMWF ERA-5 reanalysis (0.25° × 0.25°) was used to define "land" and "ocean" surface types in the first instance. Grid cells with land mask values greater than 0.5 were attributed to "land", the rest to "ocean". For coordinates assigned to the "ocean" surface type, if their sea-ice fraction (SIC, from satellite retrievals) was between 15 and 85%, they were assigned to the "marginal ice zone" (MIZ) surface type; if their SIC

was above 85%, they were assigned to the "pack ice" surface type[21]. It was also important to differentiate between different types of landmasses. While still identified as "land" using the routine outlined above, landmasses at latitudes higher than 60 °S should not be lumped with those at lower latitudes as they have much lower isoprene emissions (Fig. 4c). Here we separated landmasses at latitudes north of 60 °S (temperate zone) from those below (Antarctic zone). Ice shelves (e.g., the Ross ice shelf) were considered part of Antarctic (>60°S) landmasses.

### ISOREMS fluxes

Monthly isoprene sea-to-air fluxes were calculated for December 2016, January, February and March 2017 using the ISOREMS approach, specifically the parameterisations of Eqs. 1–4[12]. Prior work[42] identified chlorophyll-a and SST as major predictors of aqueous isoprene concentrations, representing the combination of phytoplankton biomass, temperature effects on biomass-specific isoprene production[5], and taxonomic composition of phytoplankton over regional scales.

To calculate the emissions of isoprene, we first regridded monthly mean chlorophyll-A (CHLA) concentrations from MODIS-Aqua to the resolution used by ERA5. We chose to use hourly wind speed and SST from ERA5 rather than the monthly means of squared wind speed used by Rodríguez-Ros et al.[12] and so applied a linear interpolation of the monthly CHLA fields to generate data on hourly time points. This CHLA data was then used alongside ERA5 hourly SST data to calculate aqueous isoprene concentration, ISO (Eq. 1 in Rodríguez-Ros et al.[12], mean value). The ISO field was then used to calculate isoprene emissions, $F_{ISO}$, following Eq. 2 of Rodríguez-Ros et al.[12], which also used ERA5 wind speed and SST. The resulting isoprene emissions were then scaled to give the units of $kg\ m^{-2}\ s^{-1}$ and averaged to give monthly mean values as required by UKESM1, and conservatively regridded to UKESM1 N96 resolution. We only considered the region from 90°S to 30°S in this study.

Emissions were applied as a prescribed field (i.e., simulated model meteorology did not affect emission rates). As is standard in UKESM1, for each grid cell, emissions were linearly interpolated between monthly mean values with a fixed period of 5 days before being stepped to the next value. When a diurnal cycle for marine isoprene emissions was not applied, emissions were constant throughout the day. In the simulations where diurnal cycles were applied to marine isoprene emissions, the same stepping approach was applied but emissions varied between 0 at night and a maximum during daylight hours.

For runs where marine isoprene emissions were scaled (e.g., TI_MI_20x), a universal scaling was applied to all grid cells and time points (i.e., marine isoprene emissions in TI_MI_20x were 20x higher at any time point and location than in TI_MI_MEAN).

In the case of TI_MI_MEAN_D_sink and TI_MI_MEAN_D_sink_100x, the emissions of the dummy species acting as an OH sink were equal to and 100 times those of the marine isoprene emissions in TI_MI_MEAN respectively. Like the marine isoprene emissions in TI_MI_MEAN, the dummy emissions had no diurnal scaling. The dummy species reacted with OH, generating only $CO_2$ to prevent any oxidant regeneration. The rate constant was set as half that of OH + isoprene[56] (given the high reactivity of isoprene with OH), with the same temperature dependence:

$$dummy + OH \rightarrow CO_2; k = 1.35 \times 10^{-11} e^{390/T} \qquad (2)$$

### UKESM model

All model runs were performed using the United Kingdom Earth System Model v1.0 (UKESM1) in atmosphere-only (AMIP) setup, run at a horizontal resolution of $1.25° \times 1.875°$ with 85 vertical levels up to 85 km[57], and the GLOMAP-mode aerosol scheme, which simulates sulfate, sea salt, black carbon, organic matter, and dust but does not simulate currently nitrate aerosol[58]. In this setup, the inert chemical tracer Sec_Org, which condenses irreversibly onto existing aerosol, was produced at a 26% yield solely from reactions of α-pinene and β-pinene with $O_3$, OH, and $NO_3$ with the enhanced yield applied to account for a lack of SOA formation from isoprene or anthropogenic species[58]. All runs used the CRI-Strat 2 chemistry scheme,[59] which uses the updated isoprene of Jenkin et al.[60]

Temperature and horizontal wind fields were nudged[61] in all model runs to atmospheric reanalyses from ECMWF[62] to constrain the simulations to consistent meteorology, thus preventing diverging meteorology from adding to the differences resulting from the chemical mechanisms and replicating the atmospheric conditions experienced when the observations were recorded as closely as possible. Nudging only occurred above ~1200 m in altitude, and thus the majority of the planetary boundary layer was not nudged. The model runs were atmosphere-only runs with prescribed sea surface temperatures (SSTs). $CO_2$ is not emitted but set to a constant field, while methane, CFCs, and $N_2O$ are prescribed with constant lower boundary conditions, all at 2014 levels[63].

The emissions used in this study are the same as those developed for the Coupled-Model Intercomparison Project 6 (CMIP6)[64]. Anthropogenic and biomass burning emissions data for CMIP6 are from the Community Emissions Data System (CEDS), as described by Hoesly et al.[65]. 2014 timeslice emissions were used for anthropogenic and biomass burning emissions. Oceanic emissions were from the POET 1990 data set[66], and all biogenic emissions, including terrestrial isoprene, were based on 2001–2010 climatologies from Model of Emissions of Gases and Aerosols from Nature under the Monitoring Atmospheric Composition and Climate project (MEGAN-MACC) (MEGAN) version 2.1[1]. All runs used the same anthropogenic, biomass burning and biogenic emissions, with the only difference being the addition of emissions of marine isoprene and the dummy species used as an OH sink in certain runs. These two sets of emissions are discussed in more detail in the ISOREMS flux sections.

### Model-observation comparison along ship track

For each hourly observation data point, we extracted the contemporaneous modelled concentration for the grid cell whose centre coordinate displayed the smallest difference in latitude and longitude to the coordinate of the observational data point.

### Diel cycle

Local solar times were calculated by adding a factor of 4 minutes per degree longitude (−180° to 180°) to UTC time[67]. To ensure the diel cycle was representative of marine sources alone, the isoprene data points included in the calculations were those for which the ocean contribution along the back trajectory was greater than 99% and the terrestrial isoprene from TI_BASE runs was lower than 1 ppt. To reduce the impact of spikes and short-term variability, an additional set of diel cycles (Fig. S17) was produced in which every isoprene concentration was normalised to the corresponding daily mean concentration following the procedure described in Wohl et al.[10] Night-time was defined as periods with solar irradiance <1 W m$^{-2}$ (see Fig. S21).

## Data availability

All the data from the ACE campaign can be found in the ACE Zenodo online repository[39] at https://zenodo.org/communities/spi-ace/ (last accessed: 15 August 2023). Ambient isoprene mixing ratios are available from Bolas et al.[68] https://zenodo.org/records/5674685. The URLs for other datasets used in this work are provided within the text and references for the Methods section. All model data are freely available at the Zenodo repository https://doi.org/10.5281/zenodo.8184979 and https://zenodo.org/records/8184980.

## Code availability

Due to intellectual property right restrictions, we cannot provide either the source code or documentation papers for the UM. The Met Office United Model is available for use under licence. A number of research organisations and national meteorological services use the UM in collaboration with the UK Met Office to undertake atmospheric process research, produce forecasts, develop the UM code, and build and evaluate Earth system models. For further information on how to apply for a licence, see https://www.metoffice.gov.uk/research/approach/modeling-systems/unified-model (last accessed: 20 Dec 2023).

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

## Acknowledgements

The Antarctic Circumnavigation Expedition was funded by the Swiss Polar Institute and Ferring Pharmaceuticals. A. Baccarini was supported by the SNSF Grant No. 200021_169090. J. Schmale holds the Ingvar Kamprad Chair for Extreme Environments Research. R. Simó holds a European Research Council Advanced Grant (ERC-2018-AdG #834162). The ICM-CSIC is supported by a "Severo Ochoa" Centre of Excellence grant (CEX2019-000928-S) from the Spanish government. This work used Monsoon2, a collaborative high-performance computing facility funded by the Met Office and the Natural Environment Research Council. This work used Joint Analysis System Meeting Infrastructure Needs (JASMIN), the UK collaborative data analysis facility. J.W.'s work was supported by the United Kingdom Research and Innovation (UKRI) Future Leaders Fellowship Programme awarded to Dr Maria Val Martin (MR/T019867/1). V.F. and N.R.P.H. were supported in the analysis of the data by UKRI NERC project Southern Ocean Clouds (NE/T006366/1). V.F. and J.W. acknowledge the map sources used throughout this paper: ESRI and NOAA (NOAA National Centers for Environmental Information. 2022: ETOPO 2022 15 Arc-Second Global Relief Model. NOAA National Centers for Environmental Information. DOI: 10.25921/fd45-gt74). V.F. is grateful to Dr David M. Rowley for initial discussions on the interpretation of the isoprene measurements.

## Author contributions

C.G.B., A.D.R., and N.R.P.H. developed and prepared the iDirac for the measurements of atmospheric isoprene during ACE. J.S. led the atmospheric measurements on ACE. F.T., J.S. and A.B. operated the iDirac during ACE. V.F., C.B.G. and R.L.J. processed the isoprene data collected by the iDirac. R.S., P.R.R. and P.C.G. performed measurements of seawater isoprene and chlorophyll-a during ACE. P.R.R., R.S., M.G. and P.C.G. provided data and code on chlorophyll-a and isoprene fluxes (ISOREMS) from ACE. V.F. performed the back-trajectory analysis. J.W.

generated the emission fields of marine isoprene, set up and executed the UKESM simulations, and compared the output to observational data. V.F. and J.W. led the overall analysis and interpretation of the data collected, and drafted the original manuscript. V.F., J.W., C.G.B., A.D.R., F.T., P.R.R., P.C.G., A.B., R.L.J., M.G., R.S., J.S. and N.R.P.H. contributed to discussing, commenting and revising the manuscript.

## Competing interests

The authors declare no competing interests.
