## [Peer Review File · Nature Communications]

Atmospheric isoprene measurements reveal larger than expected Southern Ocean emissionsREVIEWER COMMENTS

Reviewer #1 (Remarks to the Author):

The manuscript reports observations of isoprene in the remote Southern Ocean marine atmosphere with particularly significant concentrations originating from the Antarctic marginal ice zone. Through model-observation comparisons the authors claim current estimates used in model isoprene fluxes result in a >20x underestimation of atmospheric isoprene in this region, implying a substantial missing source. The authors propose that production via photochemistry at the sea surface in addition to marine biological production could explain the concentrations and profile observed. Furthermore, the implications of how these higher isoprene emissions could significantly impact atmospheric composition in the Southern Ocean region in relation to oxidising capacity were explored.

As acknowledged and referenced in the manuscript these are not the first observations of the occurrence of isoprene in the Southern Ocean atmosphere nor is this the first study to identify the discrepancy between top-down and bottom-up estimates of oceanic isoprene emissions pointing to an additional photochemical source of isoprene. However, this work is a significant contribution to the available in situ measurements in this sparsely sampled region and provides a good exploration of potential sources of current model biases and potential implications for atmospheric chemistry and composition in this region. Overall, the work will be of significant interest to a wide audience in the field of ocean-atmosphere interactions and their significance to climate. The authors could consider adding comments on potential implications of the recent observed rapid changes in the Antarctic MIZ with record low sea ice in 2023.

The measurement, modelling and analysis approaches taken are robust and generally well described. Given marine biogenic emissions of isoprene have been shown to be species and nutrient dependent, the manuscript would be strengthened by a better exploration of the nature of the phytoplankton blooms and BGC conditions encountered during the voyage (or typical of this region) and how these compare to assumptions made in the model emission estimates. The manuscript is suitable for publication with some suggested minor edits:

- 1) Line 25 “ Marine emissions are poorly constrained...” loss processes in the ocean are also poorly understood so suggest use of term fluxes instead of emissions here. Also suggest use semi- colon instead of colon after “long-term measurements”
- 2) Line 28 “The highest concentrations originated in the marginal ice zone...” report the range of values considered highest.
- 3) Lines 32 – 34 “ Increased isoprene emissions would lead to” suggest rephrase “The model presented here suggests such an increase in isoprene emissions would lead to” Or similar
- 4) Introduction paragraph 4 (Lines 69-88) – suggest rather than characterise SO as a proxy for pre-industrial conditions, focus on it as a source of understanding present natural chemistry-climate processes in the background atmosphere.
- 5) Results paragraph 1 could be improved by including latitudinal ranges of each leg.
- 6) Fig 3 – text in these plots is too small.
- 7) To highlight the urgency in getting a better baseline understanding of processes occurring in this region in the discussion of future directions it would be worth highlighting the rapid changes that have

been observed in the MIZ particularly this year.

8) Line 315 – additional reference for glyoxal, methyl glyoxal from marine regions is Lawson et al 2015 doi:10.5194/acp-15-223-2015

9) Line 320 – 323 – here it needs to be clearer the scope of region and perhaps season the results conclusions presented pertain to.

10) Line 373 – 380 – it could be of interest to comment on whether the latitudinal gradients of marine isoprene align with those reported for biologically derived aerosols in this region (eg Humphries et al 2021. <https://doi.org/10.5194/acp-21-12757-2021>)

11) Line 383 - One would expect measurable losses of isoprene within the heated stainless steel tube (eg Goldstein et al 1995 Automated in-situ monitoring of atmospheric non-methane hydrocarbon concentrations and gradients, J. Atmos Chem.) The sample flow rate (inlet residence time), inlet temperature and an estimate of losses should be commented on here.

12) Figure S.5 Header text is too small.

13) Consider reducing the number of figures in Supplement by overlaying time series presented in Figs S.6 – S.9

Reviewer #2 (Remarks to the Author):

The study "Atmospheric isoprene measurements reveal larger-than-expected Southern Ocean emissions" presented by Ferracci et al. is both timely and relevant, addressing a critical knowledge gap in our understanding of marine isoprene emissions and their potential implications for atmospheric chemistry and climate. The research seeks to elucidate the contribution of the Southern Ocean to isoprene emissions, with particular emphasis on the marginal ice zone (MIZ) in the Ross and Amundsen seas. The study further delves into the discrepancies between observed isoprene concentrations and existing model estimates, highlighting the necessity of a daytime source of isoprene to reconcile these differences.

One of the strengths of this manuscript lies in the extensive and well-executed data collection and analysis. The use of long-term measurements spanning four months during the summertime offers valuable insights into the spatial and temporal variability of atmospheric isoprene over the Southern Ocean. The observation that the MIZ serves as a significant source of isoprene at high latitudes is a noteworthy discovery and carries important implications for our understanding of isoprene cycling in remote marine environments. Particularly, the implications of increased isoprene emissions on the hydroxyl radical concentrations in the Southern Ocean are significant, as they have the potential to influence atmospheric oxidation processes in this remote region.

Overall, the manuscript exhibits a strong foundation of scientific rigor and presents valuable insights into the field of atmospheric science, with direct relevance to both climate modeling and our understanding of atmospheric composition in remote environments.

In conclusion, I believe that the manuscript represents a valuable contribution to the field and has the potential to be a significant publication in Nature Communications after revising the manuscript according to my comments below.

General Comments:

The authors give and discuss mixing ratios in the text mostly in ppt, but in the figures the scales are often given in ppb. For better readability, I suggest changing the Figures to ppt values and consistently discussing ppts in the text.

The authors present and discuss several time series of measurement data. Did the authors conduct any kind of correlation analysis for these time series? E.g. does the isoprene signal show a significant correlation at certain times to chlorophyll-a or other variables (Figures 1b, 3, and 4)? Maybe also with a certain lag phase? Correlation values or scatter plots would make potential correlations much easier visible to the readers. Visually comparing time series is a tedious task and prone to false interpretations. It is really intriguing to see the good agreement of the acquired isoprene data with the TI_MI_20x_D model. However, for the reader it is difficult to understand where the scaling factor 20 comes from and how it was determined. Was this factor chosen via trial and error? Or was there some kind of algorithm used to find this scaling factor? This needs to be discussed.

Specific Comments:

Figure 3: The color code for panel e & f should be changed, as values close to 1 cannot be seen because of the white background.

Lines 259ff: It has been proposed that surfactants are especially enriched in the SML a few days after blooms of phytoplankton (i.e. when cells are dying). Would it be possible for the authors to further support this hypothesis by showing a certain delay in isoprene maxima to chlorophyll-a maxima? (see also e.g. O'Dowd et al. 2015, DOI: <https://doi.org/10.1038/srep14883>.)

Similar observations of interfacial photochemistry at the ocean surface have already been made in the Arctic Ocean. How do the measurements of the ACE campaign compare to these studies? (see e.g. Mungall et al. 2017, DOI: <https://doi.org/10.1073/pnas.1620571114>; Kim et al. 2017, DOI: <https://doi.org/10.1002/2017GL072975>)

Line 264: There is a typo in the reference. I guess it should read Brüggemann et al.

Figure 4, panels a and b: Is there a reason why the diurnal plots show only 23 hours instead of 24? As a day on earth has roughly 24 hours, this is confusing without further explanation. Also in the SI, Fig. S14-S18 and S20 show only 23 hours per day.

Reviewer #3 (Remarks to the Author):

The manuscript „Atmospheric isoprene measurements reveal larger-than-expected Southern Ocean emissions” the authors V. Ferracci & J. Weber et al., report about atmospheric isoprene measurements from Dec 2016 to Mar 2017 in the Southern Ocean region. With the help of modeled oceanic isoprene

concentrations, which are partly satellite based, the authors clearly show that biogenically produced isoprene in the surface ocean alone and the subsequent emissions to the atmosphere cannot explain measured atmospheric isoprene mixing ratios (emission to be a factor of 20 too low). A variety of different model set-ups carried out in this study lead to the conclusion that an additional light dependent daytime source of isoprene emissions, potentially in the SML, is needed to explain atmospheric mixing ratios. Testing these needed isoprene emissions in UKSEM1 results in significant decrease of OH radical mixing ratios which potentially affects the oxidative capacity of the atmosphere over the Southern Ocean.

Presentation of the data and the methodological approach clearly demonstrate a robust and well-designed study, which is reported in this manuscript. It is great to see that other potential sources, which would explain higher isoprene emissions, are ruled out, by testing different model-setups. Everything leads to the assumption that the SML might play a prominent role to be responsible for the mismatch between current bottom-up and top-down approaches. However, this is also my biggest concern about the novelty as well as the scientific result within the scope of this journal.

Isoprene emissions being a potential candidate to close the gap between bottom up and top down approaches is not new. Results from field or model campaigns from other oceanic regions other than the Southern Ocean already indicated the mismatch (e.g. Booge et al., 2016; Conte et al., 2020; Zhang et al., 2022). I totally agree, that these studies did not rule out other sources by different model approaches as this study does, however, the data in this manuscript does not directly prove the SML to be the “missing source”, which then would stand out compared to other publications.

The other prominent result in this manuscript are the implications for the oxidative capacity of the atmosphere using tuned (and needed) isoprene emissions to account for the atmospheric mixing ratios. It is important to see, that much stronger emissions of isoprene (x20) have a significant impact on OH levels over the Southern Ocean (2% reduction) during DJFM. However, looking on yearly averages the influence of isoprene emissions becomes less strong, as biogenic isoprene emissions (due to low primary production) as well as potential SML emissions (due to higher winds) are much more reduced during times of less light (Brüggemann et al., 2018; Zhou et al., 2022). This will dampen the overall influence on OH levels and should be mentioned but also tested for significance in the manuscript.

Overall, I suggest to publish this important work, based on a very comprehensive dataset, in another journal, in order to better match the results of this study with the scope of the journal.

References

- Booge, D., Marandino, C. A., Schlundt, C., Palmer, P. I., Schlundt, M., Atlas, E. L., Bracher, A., Saltzman, E. S., and Wallace, D. W. R.: Can simple models predict large-scale surface ocean isoprene concentrations?, *Atmos. Chem. Phys.*, 16, 11807–11821, <https://doi.org/10.5194/acp-16-11807-2016>, 2016.
- Brüggemann, M., Hayeck, N. & George, C. Interfacial photochemistry at the ocean surface is a global source of organic vapors and aerosols. *Nat Commun* 9, 2101 (2018). <https://doi.org/10.1038/s41467-018-04528-7>
- Conte, L., Szopa, S., Aumont, O., Gros, V., & Bopp, L. (2020). Sources and sinks of isoprene in the global

open ocean: Simulated patterns and emissions to the atmosphere. *Journal of Geophysical Research: Oceans*, 125, e2019JC015946. <https://doi.org/10.1029/2019JC015946>

Zhang, W., Gu, D. Geostationary satellite reveals increasing marine isoprene emissions in the center of the equatorial Pacific Ocean. *npj Clim Atmos Sci* 5, 83 (2022). <https://doi.org/10.1038/s41612-022-00311-0>

Zhou, L., Booge, D., Zhang, M., and Marandino, C. A.: Winter season Southern Ocean distributions of climate-relevant trace gases, *Biogeosciences*, 19, 5021–5040, <https://doi.org/10.5194/bg-19-5021-2022>, 2022.

Reviewer #1

The manuscript reports observations of isoprene in the remote Southern Ocean marine atmosphere with particularly significant concentrations originating from the Antarctic marginal ice zone. Through model-observation comparisons the authors claim current estimates used in model isoprene fluxes result in a >20x underestimation of atmospheric isoprene in this region, implying a substantial missing source. The authors propose that production via photochemistry at the sea surface in addition to marine biological production could explain the concentrations and profile observed. Furthermore, the implications of how these higher isoprene emissions could significantly impact atmospheric composition in the Southern Ocean region in relation to oxidising capacity were explored.

We thank the reviewer for their comments.

As acknowledged and referenced in the manuscript these are not the first observations of the occurrence of isoprene in the Southern Ocean atmosphere nor is this the first study to identify the discrepancy between top-down and bottom-up estimates of oceanic isoprene emissions pointing to an additional photochemical source of isoprene. However, this work is a significant contribution to the available in situ measurements in this sparsely sampled region and provides a good exploration of potential sources of current model biases and potential implications for atmospheric chemistry and composition in this region. Overall, the work will be of significant interest to a wide audience in the field of ocean-atmosphere interactions and their significance to climate. The authors could consider adding comments on potential implications of the recent observed rapid changes in the Antarctic MIZ with record low sea ice in 2023.

We thank the reviewer for this and have added text discussing the potential impact of changes in the MIZ at the end of the Discussion section where we commented on the simulated future changes to chlorophyll-a and SSTs as we believe this the most suitable place.

Future changes to marine isoprene emissions will depend on a complex mix of factors including chlorophyll-a concentrations (predicted to increase under high (SSP3-7.0; +5.8%) and low (SSP1-2.6; +2.9%) warming scenarios in the SO, Fig S2049), sea-ice dynamics, SSTs and surface winds. Significant fluctuations in sea ice extent with a general decreasing trend (e.g., record low in 2023) are likely to have important implications for the MIZ and related isoprene emissions, and this work provides a baseline for this rapidly changing environment against which future observations and models can be compared. These rapid changes indicate that any further measurements of baseline conditions have to be made urgently.

The measurement, modelling and analysis approaches taken are robust and generally well described. Given marine biogenic emissions of isoprene have been shown to be species and nutrient dependent, the manuscript would be strengthened by a better exploration of the nature

of the phytoplankton blooms and BGC conditions encountered during the voyage (or typical of this region) and how these compare to assumptions made in the model emission estimates.

Prior work (Rodriguez-Ros, GRL, 2020) has explored the links between phytoplankton communities and seawater isoprene, and found that, on basin/regional ocean scales, chlorophyll-a and SST explained an important fraction of isoprene variance. Underlying the combination of these two predictors there is likely a combination of phytoplankton biomass (chlorophyll-a), the effect of SST on specific isoprene production (i.e., production per unit of biomass; Simó et al., 2022), and variations in the taxonomic composition of phytoplankton. This is why the parameterisation used in this study to calculate seawater isoprene concentrations and subsequently the emissions of isoprene to the atmosphere only uses chlorophyll-a and SSTs as inputs.

To clarify this we have added the following text to the “ISOREMS Fluxes” subsection of the Methods.

Prior work⁴³ identified chlorophyll-a and SST as major predictors of aqueous isoprene concentrations, representing the combination of phytoplankton biomass, temperature effects on biomass-specific isoprene production⁵, and taxonomic composition of phytoplankton over regional scales.

The manuscript is suitable for publication with some suggested minor edits:

1) Line 25 “Marine emissions are poorly constrained...” loss processes in the ocean are also poorly understood so suggest use of term fluxes instead of emissions here. Also suggest use semi- colon instead of colon after “long-term measurements”

We have amended this.

2) Line 28 “The highest concentrations originated in the marginal ice zone...” report the range of values considered highest.

We have added this.

3) Lines 32 – 34 “Increased isoprene emissions would lead to” suggest rephrase “The model presented here suggests such an increase in isoprene emissions would lead to” Or similar

We have rephrased this.

4) Introduction paragraph 4 (Lines 69-88) – suggest rather than characterise SO as a proxy for pre-industrial conditions, focus on it as a source of understanding present natural chemistry-climate processes in the background atmosphere.

We have edited that paragraph to reflect the fact that an improved understanding of atmospheric composition in the SO is key to both processes in the present-day background atmosphere as well as pre-industrial conditions since both are, by definition, largely free from

anthropogenic influence and so are not mutually exclusive. The paragraph now reads as follows:

The SO is uniquely important to understanding natural chemistry-climate processes in present-day pristine environments and it is often used as a proxy for pre-industrial environmental conditions.

5) *Results paragraph 1 could be improved by including latitudinal ranges of each leg.*
These have been added to the text.

6) *Fig 3 – text in these plots is too small.*
We have increased the size of the text in these plots.

7) *To highlight the urgency in getting a better baseline understanding of processes occurring in this region in the discussion of future directions it would be worth highlighting the rapid changes that have been observed in the MIZ particularly this year.*

We have combined our response to this point with the one to the comment above about the role of the MIZ, and have copied the changes we made to the manuscript below:

Future changes to marine isoprene emissions will depend on a complex mix of factors including chlorophyll-a concentrations (predicted to increase under high (SSP3-7.0; +5.8%) and low (SSP1-2.6; +2.9%) warming scenarios in the SO, Fig S2049), sea-ice dynamics, SSTs and surface winds. Significant fluctuations in sea ice extent with a general decreasing trend (e.g., record low in 2023) are likely to have important implications for the MIZ and related isoprene emissions, and this work provides a baseline for this rapidly changing environment against which future observations and models can be compared. These rapid changes indicate that any further measurements of baseline conditions have to be made urgently.

8) *Line 315 – additional reference for glyoxal, methyl glyoxal from marine regions is Lawson et al 2015 doi:10.5194/acp-15-223-2015*
This was added to the text (reference 33 in the revised manuscript)

9) *Line 320 – 323 – here it needs to be clearer the scope of region and perhaps season the results conclusions presented pertain to.*
We have amended the paragraph as follows:

Our work addresses the discrepancy between bottom-up and top-down estimates of marine isoprene emissions, indicating that seawater-derived (bottom-up) fluxes are insufficient to explain the concentrations of atmospheric isoprene in the SO during the austral summer, and that fluxes at least 20 times larger are needed.

10) *Line 373 – 380 – it could be of interest to comment on whether the latitudinal gradients of*

marine isoprene align with those reported for biologically derived aerosols in this region (eg Humphries et al 2021. <https://acp.copernicus.org/articles/21/12757/2021/>)

The latitudinal gradient of ambient isoprene reported in our manuscript does exhibit some degree of similarity with that of biologically derived aerosols from Humphreys et al., with higher abundances at the two extremes of the latitudinal range under consideration (40-70°S), as shown below.

However, while we recognise there may be similarities in the two latitudinal gradients, we also do acknowledge that the formation of biologically derived aerosols would be primarily driven by other factors with their latitudinal gradients, such as temperature and the emissions of other (more efficient) secondary aerosol precursors (notably, DMS). We can envisage future studies in which implementations of marine isoprene chemistry in models can be assessed in terms of its contributions to the aerosol burden. We have added the following statement in the “Future direction” section of the manuscript:

Moreover, the hygroscopicity of cloud condensation nuclei in the SO has been found to be much lower than expected, pointing towards a significant role of organic aerosol species^{13,35}. The higher-than-expected isoprene emissions in the region are likely one explanation. Model simulations where isoprene is coupled explicitly to the organic aerosol scheme could be used to probe this in future work, with detailed comparison to observations.

11) Line 383 - One would expect measurable losses of isoprene within the heated stainless steel tube (eg Goldstein et al 1995 Automated in-situ monitoring of atmospheric non-methane hydrocarbon concentrations and gradients, J. Atmos Chem.) The sample flow rate (inlet residence time), inlet temperature and an estimate of losses should be commented on here.

We have added the details requested by the reviewer to the relevant section of the Methods:

Ambient air was sampled from an inlet mounted on a container located on the second deck of the vessel at a height of approximately 15 m above the ocean surface.³⁸ The main inlet consisted of a heated-2-m, 1" OD stainless steel tube (1" outer diameter) heated to 20 °C.

Ambient air was sampled at 13 L min^{-1} , resulting in a residence time in the main inlet of $< 3 \text{ s}$. Mixing ratios of ambient isoprene were monitored using the iDirac, an autonomous portable gas chromatograph coupled with photoionisation detection (GC-PID)⁴⁰. The iDirac sampled from the main air inlet via a $\sim 50\text{-cm}$, $1/16''\text{OD}$, $0.04'' \text{ ID}$ SilcoNert2000-treated stainless steel tube, at a rate of 20 mL min^{-1} , resulting in a residence time of $1\text{-}2 \text{ s}$. The use of passivated surfaces minimised surface losses.

We also note that the Goldstein et al. reference mentioned by the reviewer utilised a rather more complex sampling system, which included very long non-heated sampling lines ($3/8'' \text{ OD}$, 24 and 29 m length) and a cryotrap. The sampling system used for the ACE cruise, by contrast, used a short wide-bore tube ($1'' \text{ OD}$, 2 m length) heated to 20°C (i.e., above ambient temperature for the vast majority of the field campaign) and a passivated thinner tube ($1/16''\text{OD}$ $0.04'' \text{ ID}$ SilcoNert2000-treated stainless steel tube), with total residence times $< 5 \text{ s}$, for which losses would be negligible.

12) *Figure S.5 Header text is too small.*

We have increased the size of the text in these plots.

13) *Consider reducing the number of figures in Supplement by overlaying time series presented in Figs S.6 – S.9*

We did try plotting the ship measurement data, UKESM data and ERA5 data overlaid on a single panel but found this reduced the clarity of plot and made it much harder to compare the data from the different sources. However, in response to the reviewer's comment, we have added horizontal lines at -5 , 5 and 15 m s^{-1} for Figs S8 and S9 to make comparison of the timeseries easier.

Reviewer #2 (Remarks to the Author):

The study "Atmospheric isoprene measurements reveal larger-than-expected Southern Ocean emissions" presented by Ferracci et al. is both timely and relevant, addressing a critical knowledge gap in our understanding of marine isoprene emissions and their potential implications for atmospheric chemistry and climate. The research seeks to elucidate the contribution of the Southern Ocean to isoprene emissions, with particular emphasis on the marginal ice zone (MIZ) in the Ross and Amundsen seas. The study further delves into the discrepancies between observed isoprene concentrations and existing model estimates, highlighting the necessity of a daytime source of isoprene to reconcile these differences.

One of the strengths of this manuscript lies in the extensive and well-executed data collection and analysis. The use of long-term measurements spanning four months during the summertime offers valuable insights into the spatial and temporal variability of atmospheric isoprene over the Southern Ocean. The observation that the MIZ serves as a significant source of isoprene at high latitudes is a noteworthy discovery and carries important implications for our understanding of isoprene cycling in remote marine environments. Particularly, the implications of increased isoprene emissions on the hydroxyl radical concentrations in the Southern Ocean are significant, as they have the potential to influence atmospheric oxidation processes in this remote region.

Overall, the manuscript exhibits a strong foundation of scientific rigor and presents valuable insights into the field of atmospheric science, with direct relevance to both climate modeling and our understanding of atmospheric composition in remote environments. In conclusion, I believe that the manuscript represents a valuable contribution to the field and has the potential to be a significant publication in Nature Communications after revising the manuscript according to my comments below.

We thank the reviewer for their comments and are pleased that they find our analysis and data collection to be well-executed.

General Comments:

The authors give and discuss mixing ratios in the text mostly in ppt, but in the figures the scales are often given in ppb. For better readability, I suggest changing the Figures to ppt values and consistently discussing ppts in the text.

We have changed the units in all relevant plots to ppt.

The authors present and discuss several time series of measurement data. Did the authors conduct any kind of correlation analysis for these time series? E.g. does the isoprene signal show a significant correlation at certain times to chlorophyll-a or other variables (Figures 1b, 3, and 4)? Maybe also with a certain lag phase? Correlation values or scatter plots would make

potential correlations much easier visible to the readers. Visually comparing time series is a tedious task and prone to false interpretations.

We looked at correlation metrics between observed isoprene and chlorophyll-a using different approaches (table below), e.g. chlorophyll-a along the ship track, averaged chlorophyll-a along the back trajectory, chlorophyll-a with a time lag compared to isoprene observations (related to a point raised by the reviewer later on).

Chlorophyll-a	Data subset	R²	p-value
MODIS 8-day composite, no lag, along cruise track only	All ACE	0.05	2e-09
	Leg 1	0.45	< 2.2e-16
	Leg 2	0.005	0.17
	Leg 3	0.002	0.68
	North of 50°S	0.20	< 2.2e-16
	South of 50°S	0.007	0.12
MODIS 8-day composite, 8-day lag, along cruise track only	All ACE	0.13	< 2.2e-16
	Leg 1	0.46	< 2.2e-16
	Leg 2	0.005	0.19
	Leg 3	0.001	0.77
	North of 50°S	0.20	2e-15
	South of 50°S	0.02	0.03
MODIS 8-day composite, no lag, average along adjusted back-trajectory	All ACE	0.02	3e-05
	Leg 1	0.25	< 2.2e-16
	Leg 2	0.004	0.15
	Leg 3	0.003	0.64
	North of 50°S	0.10	2e-11
	South of 50°S	0.007	0.07

The data in the table indicates that ambient isoprene exhibits the strongest correlation to chlorophyll-a during Leg 1 under all scenarios, and typically data north of 50°S exhibit higher

correlation than data south of 50°S. Averaging along the back-trajectory does not improve the correlations, whereas introducing a lag time leads to a marginal improvement in the correlation. This might indicate that chlorophyll-a can indeed be a proxy for local sources, which dominate when isoprene lifetime is short (*i.e.*, during Leg 1 and north of 50°S), but even in the best of cases correlation remains low ($R^2 < 0.5$). We also note that terrestrial emissions influence observed isoprene in the vicinity of land at the start of both legs 1 and 2. Poor correlation in Leg 2 in particular would indicate that chlorophyll-a is a poor proxy in this section of the campaign, which would be consistent with a stronger SML source in this region. We attribute the poor correlation during Leg 3 to the paucity of data available for that section of the campaign (< 3 days).

We have included the Table above as Table S1 in the Supplementary Information, along with a description of the analysis carried out. This analysis is also used to support the statements in the Results section as follows:

At mid-latitudes (30-50 °S), particularly during Leg 1, atmospheric isoprene and chlorophyll-a (Fig 1c) followed similar decreasing trends as the ship travelled away from the coast (see Suppl. Mat. and Table S1). However, the two decoupled in Leg 2, especially at high latitudes (> 70 °S), where the elevated isoprene mixing ratios and baseline enhancement in Leg 2 did not correspond to high dissolved isoprene or chlorophyll-a concentrations (see Table S1).

It is really intriguing to see the good agreement of the acquired isoprene data with the TI_MI_20x_D model. However, for the reader it is difficult to understand where the scaling factor 20 comes from and how it was determined. Was this factor chosen via trial and error? Or was there some kind of algorithm used to find this scaling factor? This needs to be discussed.

The decision to scale by a factor of 20 was taken based on TI_MI_MEAN's large low bias. We acknowledge TI_MI_20x remains low-biased, but we decided that it was sufficient in demonstrating the substantial underestimate of marine isoprene fluxes in the Southern Ocean. We have amended the text in the "Model - Observation Comparisons" section of the Results as follows:

~~In an effort to address this low bias, we scaled~~ When we scale these marine isoprene emissions by a factor of 20 (TI_MI_20x), ~~This improved the model performance improves but it remained~~ low-biased by a factor of ~2...

Specific Comments:

Figure 3: The color code for panel e & f should be changed, as values close to 1 cannot be seen because of the white background.

We acknowledge the original display was harder to interpret. To make it clearer we have changed the colour of the background in the plot to grey so the white spots in Fig 3f, which indicate a model/observation ratio close to 1, are much clearer.

Lines 259ff: It has been proposed that surfactants are especially enriched in the SML a few days after blooms of phytoplankton (i.e. when cells are dying). Would it be possible for the authors to further support this hypothesis by showing a certain delay in isoprene maxima to chlorophyll-a maxima? (see also e.g. O'Dowd et al. 2015, DOI: <https://doi.org/10.1038/srep14883>.)

We looked into this by introducing an 8-day lag between chlorophyll-a and isoprene, so that we are effectively looking at the chlorophyll 8 days prior to ship's passage over a particular spot. We only see a marginal improvement in correlation between ambient isoprene and chlorophyll-a, as illustrated in the table below.

	R² (No lag in chl-a)	R² (8-day lag in chl-a)
All ACE	0.05	0.13
Leg 1	0.45	0.46
Leg 2	0.005	0.005
Leg 3	0.002	0.001
North of 50°S	0.20	0.20
South of 50°S	0.007	0.02

Of course we mainly expect the SML to be stable under the low-wind conditions encountered in Leg 2. We overlaid the lagged MODIS chlorophyll-a on Figure 1c (below, brown line) and it appears to generally follow the non-lagged MODIS chlorophyll-a (green line). It is worth noting that there are many gaps in the MODIS time series during Leg 2, especially at the most Southern latitudes, which further complicate this type of analysis.

Similar observations of interfacial photochemistry at the ocean surface have already been made in the Arctic Ocean. How do the measurements of the ACE campaign compare to these studies? (see e.g. Mungall et al. 2017, DOI: <https://doi.org/10.1073/pnas.1620571114>; Kim et al. 2017, DOI: <https://doi.org/10.1002/2017GL072975>)

It is difficult for us to compare our results to these studies due to the lack of SML sampling during ACE and limited knowledge of photochemical precursors of isoprene. As suggested in the Mungall et al. (2017) paper, the photochemical nature of the isoprene source is best indicated by the correlation between the diurnal cycles of the isoprene and solar radiation, which we have plotted below (with shaded areas indicating data below the instrument LOD):

Line 264: There is a typo in the reference. I guess it should read *Brüggemann et al.*
 We have amended this.

Figure 4, panels a and b: Is there a reason why the diurnal plots show only 23 hours instead of 24? As a day on earth has roughly 24 hours, this is confusing without further explanation. Also in the SI, Fig. S14-S18 and S20 show only 23 hours per day.

We have indexed hours starting at 0, corresponding to 00:00 local time, and finishing at 23, corresponding to 23:00 local time, thus covering the full diurnal cycle without plotting 00:00 local time twice. To make this clear we have added the following text to the caption in Figure 4:

“The 0 solar time of day corresponds to 00:00 local time and 23 corresponds to 23:00 local time.”

Reviewer #3 (Remarks to the Author):

The manuscript „Atmospheric isoprene measurements reveal larger-than-expected Southern Ocean emissions” the authors V. Ferracci & J. Weber et al., report about atmospheric isoprene measurements from Dec 2016 to Mar 2017 in the Southern Ocean region. With the help of modeled oceanic isoprene concentrations, which are partly satellite based, the authors clearly show that biogenically produced isoprene in the surface ocean alone and the subsequent emissions to the atmosphere cannot explain measured atmospheric isoprene mixing ratios (emission to be a factor of 20 too low). A variety of different model set-ups carried out in this study lead to the conclusion that an additional light dependent daytime source of isoprene emissions, potentially in the SML, is needed to explain atmospheric mixing ratios. Testing these needed isoprene emissions in UKSEM1 results in significant decrease of OH radical mixing ratios which potentially affects the oxidative capacity of the atmosphere over the Southern Ocean.

Presentation of the data and the methodological approach clearly demonstrate a robust and well-designed study, which is reported in this manuscript. It is great to see that other potential sources, which would explain higher isoprene emissions, are ruled out, by testing different model-setups. Everything leads to the assumption that the SML might play a prominent role to be responsible for the mismatch between current bottom-up and top-down approaches.

We thank the reviewer for their comments and are pleased to see they find our study robust and well-designed.

However, this is also my biggest concern about the novelty as well as the scientific result within the scope of this journal.

Isoprene emissions being a potential candidate to close the gap between bottom up and top down approaches is not new. Results from field or model campaigns from other oceanic regions other than the Southern Ocean already indicated the mismatch (e.g. Booge et al., 2016; Conte et al., 2020; Zhang et al., 2022). I totally agree, that these studies did not rule out other sources by different model approaches as this study does, however, the data in this manuscript does not directly prove the SML to be the “missing source”, which then would stand out compared to other publications.

In terms of standing out from other publications, this is the first study to combine observational data of marine isoprene and its emissions with an extensive analysis in a global CMIP6-era Earth System model. By using contemporaneous atmospheric conditions by the application of meteorological nudging, explicit atmospheric chemistry including detailed isoprene chemistry and high time frequency model output to track as closely as possible a field campaign, we build on the previous studies mentioned by the reviewer. Booge et al. (2016) used a “simple” model without explicit isoprene chemistry (an atmospheric lifetime of 1-4 hours was assumed, which we show is a significant underestimate at high latitudes) to examine atmospheric isoprene concentrations. Conte et al. (2020) used only an ocean biogeochemical model to investigate

aqueous isoprene concentrations and indeed state their new estimates “are proposed for use in atmospheric chemistry models, to better appraise the importance of oceanic BVOCs on the MBL chemistry”, which is a key focus of this current work. Zhang et al. (2022) used an offline parameterisation to calculate marine isoprene emissions based on aqueous isoprene concentrations but also did not calculate the impact on the resulting atmospheric concentration of isoprene.

In this study, we compare modelled and observed atmospheric isoprene concentrations across a whole field campaign. We demonstrate, more conclusively than past studies, the characteristics of the discrepancy between bottom-up and top-down emission approaches and the magnitude of this discrepancy. In particular, we note that a minimum 20-fold increase in emissions along with a diurnal emission cycle are needed to reconcile the model with observations, which previous studies did not identify. We also rule out many of the potential drivers of the mismatch (e.g., rapid vertical mixing or a missing sink of OH) which other studies were unable to do to the same extent, thus adding novelty.

Further, we extend our understanding of the field by identifying the atmospheric implications of including marine isoprene emissions in UKESM1. We calculate a 2% reduction in the hydroxyl radical for DJFM, rising to 14% in places, and are pleased to see Reviewer #3 agrees this is an important result. Therefore, while a definitive answer to the “missing source” is not identified, this study represents a significant progression of the field due to the constraints it places on candidates for the missing source and it will aid wider research in this area.

The other prominent result in this manuscript are the implications for the oxidative capacity of the atmosphere using tuned (and needed) isoprene emissions to account for the atmospheric mixing ratios. It is important to see, that much stronger emissions of isoprene (x20) have a significant impact on OH levels over the Southern Ocean (2% reduction) during DJFM. However, looking on yearly averages the influence of isoprene emissions becomes less strong, as biogenic isoprene emissions (due to low primary production) as well as potential SML emissions (due to higher winds) are much more reduced during times of less light (Brüggemann et al., 2018; Zhou et al., 2022). This will dampen the overall influence on OH levels and should be mentioned but also tested for significance in the manuscript.

While we agree that the DJFM period will have highest emissions of marine isoprene, we note that emissions during other parts of the year are far from negligible. The emissions developed by Rodriguez-Ros et al. (2020), on which the emissions used in this current work are based, are ~8 GgC per month in January (for 40-90°S) but even in June and July they are 2-3 GgC per month (Figure 4c of Rodriguez-Ros et al., 2020).

We also note that just as aqueous isoprene (and therefore marine isoprene emissions) decreases in the austral winter as primary productivity decreases, concentrations of OH will decrease due to reductions in primary OH production from ozone photolysis as the intensity and duration of sunlight decreases.

We still believe that focusing on the DJFM period is most appropriate due to the already demonstrated importance of constraining to observations. However, to probe this further and in response to this comment, we ran a further full year simulation using 20x marine isoprene emissions and compared the annual mean change in OH to the control run (TI_Base). We note the validity of the 20x scaling is more questionable given the absence of direct measurements of ambient isoprene beyond the DJFM period, but we found that annual surface OH concentrations decreased by 1.7% (compared to 2% for DJFM) when averaged over the same region as Fig 4d, and the pattern of such reduction is the same (newly added Fig S19, also below). To this end, we have added the following text in the “Impact on the oxidising capacity of the atmosphere” section of the manuscript:

We note that the DJFM period considered here, chosen to coincide with the ACE campaign and thus allowing us to constrain the marine isoprene emissions to observations, is the period in the year with the highest SO marine isoprene emissions (Rodriguez-Ros et al., 2020). Therefore, including marine isoprene emissions in this period will have the largest absolute impact on the hydroxyl radical and wider atmospheric composition. However, marine isoprene emissions are far from negligible in other parts of the year, even in the austral winter (2-3 GgC/month in June and July compared to 8 GgC/month in January, Rodriguez-Ros et al., 2020). The austral winter’s much lower OH concentration, due to the lower primary production from ozone photolysis given the reduced intensity and duration of sunlight, means the relative change in OH with the inclusion of marine isoprene could be comparable to that in austral summer. To probe the impact of a 20-fold scaling for the annual mean change in surface OH, we ran a further simulation with 20x marine isoprene emissions for a whole year and compared it to the annual mean of TI_base. We found that the change in annual mean OH was 1.7% (compared to 2.1% for DJFM), with a similar spatial distribution to DJFM (Fig S19), but we note the lack of observational constraints means the validity of a 20-fold scaling for the whole year is more questionable. Overall this highlights the need for further research into the atmospheric marine isoprene concentrations throughout the year in the SO so that a full understanding of their role in atmospheric composition, aerosol and climate change be established.

Fig 4d. Percentage change in OH in lowest ~150 m between TI_MI_20x and TI_base for DJFM (as in manuscript)

Fig S19. Percentage change in annual mean OH in lowest ~150 m between run with 20x marine isoprene emissions and TI_base.

In terms of the “significance”, we are not sure if the Reviewer is asking for a statistical significance test or requesting we highlight the annual variation of marine isoprene emissions as discussed above. We believe the latter has been resolved with the manuscript amendment discussed above while taking a statistical significance test over a 4-month period is not appropriate since it would conflate the month-on-month variation due to the cycle of the marine isoprene emissions (and other factors such as actinic flux) with the difference between the control and 20x marine isoprene emissions simulations. If we had multiple years of data for the control run and 20x marine isoprene emissions run (or multiple ensemble members), we could test for statistical significance on the annual mean (or specific month means). However, the computational cost of several multi-year, meteorologically nudged simulations means this was beyond the scope of the study. Even without a statistical significance test, the percentage change in OH concentrations correlates well spatially with the marine isoprene emissions, suggesting a clear link, and so we believe our conclusions to be robust.

Overall, I suggest to publish this important work, based on a very comprehensive dataset, in another journal, in order to better match the results of this study with the scope of the journal.

We are pleased to see that Reviewer #3 finds our work important and acknowledges the comprehensive dataset we have compiled. However, we do believe this study is firmly within the scope of Nature Communications. We note that Nature Communications describes itself as:

“... an open access, multidisciplinary journal dedicated to publishing high-quality research in all areas of the biological, health, physical, chemical, Earth, social, mathematical, applied, and engineering sciences. Papers published by the journal aim to represent important advances of significance to specialists within each field.” (<https://www.nature.com/ncomms/aims>)

This study brings together research from marine biology, atmospheric chemistry, earth science and earth system modelling. Thus it is a multidisciplinary study, and as such it will be of interest to researchers in all these disciplines and clearly falls within the scope of Nature Communications.

References

- Booge, D., Marandino, C. A., Schlundt, C., Palmer, P. I., Schlundt, M., Atlas, E. L., Bracher, A., Saltzman, E. S., and Wallace, D. W. R.: Can simple models predict large-scale surface ocean isoprene concentrations?, *Atmos. Chem. Phys.*, 16, 11807–11821, <https://doi.org/10.5194/acp-16-11807-2016>, 2016.
- Brüggemann, M., Hayeck, N. & George, C. Interfacial photochemistry at the ocean surface is a global source of organic vapors and aerosols. *Nat Commun* 9, 2101 (2018). <https://doi.org/10.1038/s41467-018-04528-7>
- Conte, L., Szopa, S., Aumont, O., Gros, V., & Bopp, L. (2020). Sources and sinks of isoprene in the global open ocean: Simulated patterns and emissions to the atmosphere. *Journal of Geophysical Research: Oceans*, 125, e2019JC015946. <https://doi.org/10.1029/2019JC015946>
- Zhang, W., Gu, D. Geostationary satellite reveals increasing marine isoprene emissions in the center of the equatorial Pacific Ocean. *npj Clim Atmos Sci* 5, 83 (2022). <https://doi.org/10.1038/s41612-022-00311-0>
- Zhou, L., Booge, D., Zhang, M., and Marandino, C. A.: Winter season Southern Ocean distributions of climate-relevant trace gases, *Biogeosciences*, 19, 5021–5040, <https://doi.org/10.5194/bg-19-5021-2022>, 2022.

REVIEWERS' COMMENTS

Reviewer #1 (Remarks to the Author):

The manuscript reports observations of isoprene in the remote Southern Ocean marine atmosphere with particularly significant concentrations originating from the Antarctic marginal ice zone. Through model-observation comparisons the authors claim current estimates used in model isoprene fluxes result in a >20x underestimation of atmospheric isoprene in this region, implying a substantial missing source. The authors propose that production via photochemistry at the sea surface in addition to marine biological production could explain the concentrations and profile observed. Furthermore, the implications of how these higher isoprene emissions could significantly impact atmospheric composition in the Southern Ocean region in relation to oxidising capacity were explored.

As acknowledged and referenced in the manuscript these are not the first observations of the occurrence of isoprene in the Southern Ocean atmosphere nor is this the first study to identify the discrepancy between top-down and bottom-up estimates of oceanic isoprene emissions pointing to an additional photochemical source of isoprene. However, this work is a significant contribution to the available in situ measurements in this sparsely sampled region and provides a good exploration of potential sources of current model biases and potential implications for atmospheric chemistry and composition in this region. Overall, the work will be of significant interest to a wide audience in the field of ocean-atmosphere interactions and their significance to climate. The authors have satisfactorily and to the extent possible addressed review comments in the revised manuscript. The manuscript is recommended for publication in Nature Communications.

Reviewer #2 (Remarks to the Author):

The authors have addressed all of my questions and remarks appropriately, e.g., by including a correlation analysis and revising the figures.

I suggest the manuscript for publication in Nature Communications.

Reviewer #3 (Remarks to the Author):

The revised version of the manuscript „Atmospheric isoprene measurements reveal larger-than-expected Southern Ocean emissions” by Ferracci & Weber et al., addressed the comments/suggestions from all reviewers. Additionally, the authors were convincing why Nature Communications is an appropriate journal to publish their research.

After revising the updated documents, I only have minor comments, which should be addressed before

publication.

Manuscript:

I.99: ppb does not have to be introduced anymore as only ppt is used in the manuscript. Please change to “(up to 1200 ppt)”.

I.102: Please add “Fig.S2” to “Suppl. Mat.”

I.255: Did the authors also try to normalise their data to the daily mean instead of the daily maximum? This should reduce the impact of high values (e.g. the maximum, but no outlier) on the normalisation.

I.267: There is a typo in the reference. It should read “Brüggemann”

I.289: There seems to be a mismatch of the names for the simulation in the legend of Fig.4c and the corresponding figure caption.

I.580: The text mentions “normalisation to corresponding daily mean” but figure caption of Fig.S17 says “normalisation to daily maximum”, so does the text in I.255. Please check to be consistent.

Supplement:

Fig.S1: Please indicate in the figure caption that a logarithmic scale for the y axis in (a), (b), (e) was used. This information helps to better interpret the data, especially when comparing to other variables within the same plot.

Fig.S6: y axis should contain the unit “degree” or “°”

Fig.S17: “Values have been normalised to the daily maximum isoprene (after Wohl et al., 2020)”. Wohl et al., 2020 normalised the values to the daily average isoprene concentration.

Supplementary references: Wohl et al., (2020) is missing.

Response to Reviewer Comments

We are grateful to the editor and reviewers for their comments and efforts which have helped us improve this manuscript. We have responded to each reviewers' comments below with italicised text showing the editor's and reviewers' comments and plain text showing our response. Text which has been added to the manuscript is coloured red. Original manuscript text is in blue and any text which has been removed from the manuscript is in blue and has been struck through. The locations of changes in the main text are stated.

We believe these revisions effectively address the concerns raised by the reviewer. All co-authors agree and support the submission of this revised version of our work.

Reviewer #3 (Remarks to the Author):

The revised version of the manuscript „Atmospheric isoprene measurements reveal larger-than-expected Southern Ocean emissions” by Ferracci & Weber et al., addressed the comments/suggestions from all reviewers. Additionally, the authors were convincing why Nature Communications is an appropriate journal to publish their research. After revising the updated documents, I only have minor comments, which should be addressed before publication.

We thank the reviewer for their comments.

Manuscript:

I.99: ppb does not have to be introduced anymore as only ppt is used in the manuscript. Please change to “(up to 1200 ppt)”.

Amended to:

Particularly high mixing ratios (up to 1200 ppt ~~1.2 nmol mol⁻¹, or ppb,~~) were observed at high latitudes during Leg 2

I.102: Please add “Fig.S2” to “Suppl. Mat.”

Amended

I.255: Did the authors also try to normalise their data to the daily mean instead of the daily maximum? This should reduce the impact of high values (e.g. the maximum, but no outlier) on the normalisation.

This was amended to daily mean (as opposed to daily maximum) throughout. This change did not affect the conclusion of this analysis.

I.267: There is a typo in the reference. It should read “Brüggemann”

This was amended

I.289: There seems to be a mismatch of the names for the simulation in the legend of Fig.4c and the corresponding figure caption.

This was amended as follows:

Figure 3. Observed isoprene along ship track (a-d) and modelled isoprene from simulations with (a) terrestrial isoprene emissions only (TI_base) and terrestrial isoprene emissions doubled below 30°S (TI_2x_30S), (b) terrestrial emissions with marine emissions calculated following RR's mean parameterisation (TI_MI_MEAN) and 20x these marine isoprene emissions (TI_MI_20x) and (c) terrestrial and marine emissions as TI_MI_MEAN along with an additional OH sink (details in Table 1), the emissions of which are equal to (TI_MI_MEAN_D_sink) and 100 times (TI_MI_MEAN_D_sink_100x) those of marine isoprene. Ratio of modelled isoprene emissions from (e) TI_MI_MEAN and (f) TI_MI_20x to observed isoprene. Values in (e) and (f) show *median*^{75th percentile}/_{25th percentile} of ratio across the whole campaign.

I.580: The text mentions "normalisation to corresponding daily mean" but figure caption of Fig.S17 says "normalisation to daily maximum", so does the text in I.255. Please check to be consistent.

We thank the reviewer for spotting this. It has been changed to "daily mean" throughout.

Supplement:

Fig.S1: Please indicate in the figure caption that a logarithmic scale for the y axis in (a), (b), (e) was used. This information helps to better interpret the data, especially when comparing to other variables within the same plot.

We have added this to the figure caption.

Fig.S6: y axis should contain the unit "degree" or "°"

We have added this to the y-axis label

Fig.S17: "Values have been normalised to the daily maximum isoprene (after Wohl et al., 2020)". Wohl et al., 2020 normalised the values to the daily average isoprene concentration.

We thank the reviewer for spotting this, and have replotted the figure to reflect normalisation to the daily mean (below, right) rather than the daily maximum (below, left). We are pleased that the result of the normalisation does not change the outcome, i.e. the diel cycle is still pronounced, with higher values during daytime than at night-time.

FigS17 (old version): Diel cycle of marine-originated isoprene with values have been normalised to the daily maximum isoprene

FigS17 (new version): Diel cycle of marine-originated isoprene with values have been normalised to the daily mean isoprene

Supplementary references: Wohl et al., (2020) is missing.
We have added this to the reference list.